# Lesion-Aware Reconstruction with Principal Network: Enhancing Pseudo-Label Reliability in Semi-Supervised Clinical Lesion Detection

**Shiwan DI**[1]                                                         22120036@BJTU.EDU.CN
**Jupeng LI**[1*]                                                         LIJUPENG@BJTU.EDU.CN
**Yuxuan YANG**[1]                                                       23125011@BJTU.EDU.CN
**Qian JIN**[1]                                                           24125157@BJTU.EDU.CN
**Guorui AN**[1]                                                          24120022@BJTU.EDU.CN
**Jingwen YANG**[1]                                                       25120219@BJTU.EDU.CN
**Yue WANG**[1]                                                           22120143@BJTU.EDU.CN
**Yong GUO**[1]                                                              7579@BJTU.EDU.CN
[1] *Beijing Jiaotong University, Beijing 100044, China*

**Xinyue ZHANG**[2]                                                   1810303109@BJMU.EDU.CN
**Ruohan MA**[2]                                                            KQMRH@BJMU.EDU.CN
**Gang LI**[2]                                                            KQGANG@BJMU.EDU.CN
[2] *Peking University School and Hospital of Stomatology, Beijing 100081, China*

**Editors:** Accepted for publication at MIDL 2026

## Abstract

**Purpose.** In lesion detection tasks, labeled medical data are often scarce, limiting the performance of fully supervised models. Teacher-student (TS) frameworks based on semi-supervised learning (SSL) have emerged as effective solutions to leverage unlabeled data. However, the inherent high-confidence bias of teacher networks frequently leads to the propagation of erroneous pseudo-labels, degrading the generalization ability of student networks. To address this critical issue, we propose a novel teacher-principal-student (TPS) framework.

**Methods.** The core innovation lies in introducing a principal network, which integrates lesion-aware reconstruction to filter low-quality pseudo-labels generated by the teacher network. Specifically, the principal network leverages anatomical prior knowledge and reconstruction consistency constraints to assess the reliability of teacher-generated pseudo-labels, ensuring only high-fidelity pseudo-labeled data are used for training the student network. This design fundamentally mitigates the adverse effects of the teacher prediction bias and error propagation.

**Results.** Extensive experiments on jaw lesion detection datasets demonstrate the superiority of our approach. With the same label ratio, our SSL network achieves 81.5% mAP@0.5, outperforming mainstream SSL methods by 3.0% while narrowing the performance gap with fully supervised learning to only 3.3%.

**Conclusion.** Our proposed TPS framework outperforms state-of-the-art SSL approaches in jaw lesion detection task. It not only achieves competitive performance comparable to fully supervised models but also significantly reduces reliance on labeled clinical data, providing a reliable technical solution to promote the clinical translation of lesion detection systems. Our code will be released at https://github.com/dsw847902897/Lesion-Aware-Reconstruction-with-Principal-Network.

**Keywords:** Semi-supervised learning, Enhancing pseudo-label reliability, Lesion detection, Lesion-aware reconstruction, Teacher-principal-student framework

---

* Corresponding author

## 1. Introduction

Modern medical imaging is commonly utilized in the processes of disease detection and clinical diagnosis (SHI et al., 2024; FLORES et al., 2009; SHWEEL et al., 2013). However, manual interpretation of these images is subject to the subjective factors and individual experience of the doctors, which can lead to variability in diagnosis (ABDOLALI et al., 2016; GUO et al., 2024). To overcome these limitations, deep learning-based computer-aided diagnosis (CAD) systems have been increasingly adopted in clinical practice due to their high accuracy and consistency (SUN et al., 2018; DA et al., 2019; LIU et al., 2016). These advanced systems offer a promising solution by providing more reliable and objective diagnostic support, thereby enhancing the overall quality of medical care. Nonetheless, current mainstream deep learning algorithms require a large amount of labeled data (WANG et al., 2024; SINGH et al., 2021), which is costly to obtain in clinical medicine. For locating and diagnosing lesions, the ground truth must be validated by pathological diagnosis and manually annotated and reviewed by multiple experienced clinicians, making the process time-consuming and labor-intensive. Therefore, semi-supervised learning (SSL) for detecting lesions is an effective approach to alleviate the shortage of ground truth and to make better use of unlabeled data.

However, most mainstream SSL approaches rely on the generation and utilization of pseudo-labels (REN et al., 2024; WANG et al., 2021). These methods have a significant drawback: errors in pseudo-labels can be amplified in subsequent operations, resulting in a large bias in the final network output. This issue arises from the fact that the incorrect pseudo-labels are not effectively filtered out, especially when the network predicts an incorrect result with high confidence. Particularly in jaw lesion detection tasks, during the early stages of lesion development or in lesion boundary regions where pathological features are indistinct, the teacher network tends to generate incorrect predictions with high confidence values. These high confidence errors are often not adequately screened, leading to successive optimization based on incorrect results during subsequent training of the student network. As a result, the performance of these networks becomes significantly limited. Over time, this accumulation of errors hinders the ability of the network to learn accurately, ultimately compromising the reliability and effectiveness of the model.

To address the issue of inadequate filtering of incorrect predictions in SSL, this paper proposes a teacher-principal-student (TPS) framework. Based on the classic teacher-student (TS) model, we introduce a principal network that employs lesion-aware reconstruction to evaluate and filter the predictions of the teacher network. The principal network objectively evaluates the quality of the predictions from the teacher model and filters out poorly predicted results, ensuring that the pseudo-label data used for training the student network is not influenced by the potential biases of the teacher network. Specifically, the principal network conducts image reconstruction for the lesion areas located by the teacher network, reconstructing these pathological regions into healthy tissue. These altered images, along with the original images, are input into the principal network for scoring based on contrastive learning. This model effectively measures the feature space disparity between the image pairs, and indicates more precise lesion location, signifying more accurate predictions by the teacher network. By filtering out erroneous predictions, our approach significantly improves diagnostic accuracy and reliability. Our main contributions are as follows:

- Proposed an innovative principal network for robust pseudo-label quality assessment in SSL lesion detection, integrating lesion-aware reconstruction to enhance the reliability of pseudo-labels from the teacher network, fundamentally mitigating the adverse impacts of prediction bias on the training of the student network.

- Developed a contextual attention-guided feature destruction algorithm, which adaptively disrupts lesion-related features while preserving normal anatomical integrity, laying a reliable foundation for pseudo-label quality evaluation.

- Our proposed TPS framework outperforms state-of-the-art SSL approaches in clinical lesion detection tasks, achieves competitive performance comparable to fully supervised models and significantly reduces reliance on labeled clinical data.

## 2. Related Works

Wang used a double-uncertainty method for SSL segmentation, enhancing accuracy in medical imaging with a TS model (WANG et al., 2020b). Huang compared semi- and self-supervised methods, finding MixMatch performs best under realistic conditions (HUANG et al., 2024). Studies have proposed an SSL framework for DME classification from optical coherence tomography images, enhancing accuracy with self-correction and unlabeled data (WANG et al., 2023). Peng used an SSL framework with adaptive threshold pseudo-labeling and contrastive loss to improve medical image classification (PENG et al., 2023). Qayyum proposed an SSL method for dental caries detection using self-training to enhance accuracy with limited labeled data (QAYYUM et al., 2023). Studies have shown that an SSL deep learning method can be effective for tumor pathology image analysis (JIANG et al., 2023). Wang introduced an SSL framework named FocalMix, which improves 3D medical image detection by combining focal loss with MixUp augmentation to leverage unlabeled data (WANG et al., 2020a). Similarly, Zhou proposed SSMD, an SSL medical image detection framework, improving detection with adaptive consistency and heterogeneous perturbation (ZHOU et al., 2021). Although some studies have begun to optimize the selection of pseudo-labels in SSL training, the results remain unsatisfactory. Filipiak filtered noisy pseudo-labels by setting a single confidence threshold (FILIPIAK et al., 2024). Nozarian designed category-specific foreground Intersection over Union (IoU) thresholds to classify the IoU relationship between pseudo-labels and student candidate boxes (NOZARIAN et al., 2023). The aforementioned methods filter pseudo-labels by setting thresholds, but threshold setting is subjective and lacks specificity, failing to fundamentally eliminate the impact of the teacher network errors on the student network. The principal model proposed in this paper effectively controls the quality of the teacher network's pseudo-labels, thereby mitigating this issue.

## 3. Methods

### 3.1. Overview

Our proposed TPS model structure is illustrated in Figure 1. The teacher and the student networks function as target detection networks, specifically for identifying and locating

lesions. The teacher network generates predictions that serve as pseudo-labels for the fully supervised training of the student network.

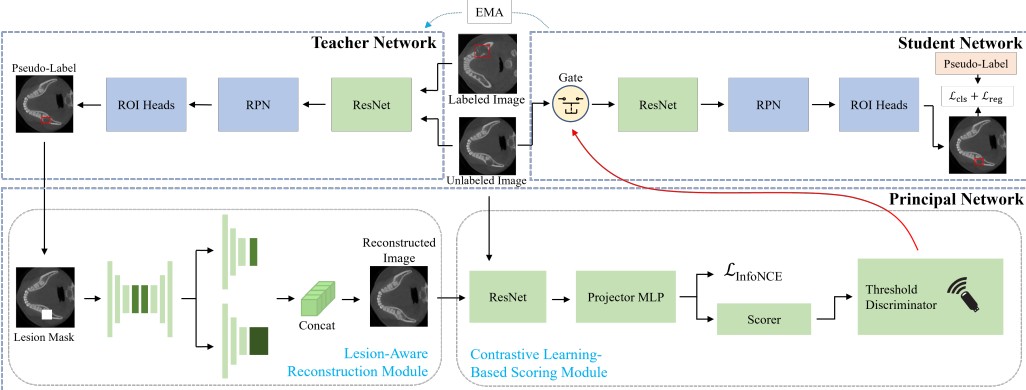

Figure 1: The structure of TPS architecture. The pre-trained teacher network generates pseudo-labels, and the student network adopts these labels based on the reliability assessment results of the principal network (via lesion-aware reconstruction and contrastive learning-based scoring module). $\mathcal{L}_{cls}$ is a classification loss and $\mathcal{L}_{reg}$ is a bounding box regression loss.

For the student network, two loss functions are adopted:

Classification Loss ($\mathcal{L}_{cls}$):

$$\mathcal{L}_{cls} = -\alpha_t (1 - p_t)^\gamma \log(p_t) \tag{1}$$

where $\alpha_t$ denotes the class-balanced weight, $p_t$ is the predicted probability of the true class, and $\gamma$ is the focusing parameter to downweight easy negative samples.

Bounding Box Regression Loss ($\mathcal{L}_{reg}$):

$$\mathcal{L}_{reg} = \frac{1}{N} \sum_{i=1}^{N} \begin{cases} 0.5x_i^2 & |x_i| \leq 1 \\ |x_i| - 0.5 & |x_i| > 1 \end{cases} \tag{2}$$

where $N$ is the number of positive samples, and $x_i = t_i - \hat{t}_i$ represents the normalized difference between the ground truth bounding box parameters ($t_i$: center coordinates, $\hat{t}_i$: the predicted parameters).

The principal network is innovatively introduced to evaluate the accuracy of these pseudo-labels to ensure the quality of predictions of the teacher network. Inaccurate predictions receive lower quality scores and are filtered out as low-quality pseudo-label data. This process ensures that the data used to train the student network is of high quality, which is crucial for improving the accuracy and reliability of the detection results. The principal network, which is the core innovation of this paper, comprises two main components: lesion-aware reconstruction module based on Contextual Attention (CAtt) (YU et al., 2018) and contrastive learning-based scoring module. The lesion-aware reconstruction module is

tasked with disrupting the features of the lesion areas identified by the teacher network and reconstructing the region into an image of healthy tissue. This change helps in evaluating how accurately the teacher network has predicted the lesions. The scoring module then compares the reconstructed image with the original image, using this comparison to assign a quality score to the predictions from the teacher network. By scoring the pseudo-labels, the principal network ensures that only high-quality labels are used for training the student network. This mechanism significantly enhances the performance of the student network by reducing the impact of the teacher network's mistakes.

### 3.2. Lesion-aware Reconstruction Module

Firstly, based on the lesion areas predicted by the teacher network, a mask corresponding to the original image is generated. Ideally, this mask will cover all the lesion areas. The resulting masked image will be used as the input for the subsequent CAtt inpainting module. The purpose of this module is to fill in the masked area based on contextual features, ensuring that contextual features of this filled area are consistent with non-lesion areas, thereby achieving our goal of disrupting features of lesion areas.

For lesion-aware reconstruction, the method adopts a generative image inpainting with contextual attention proposed in the study (YU et al., 2018). Specifically, the masked image first passes through the coarse network to generate a coarse result. This network is composed of a series of dilated convolution layers, which increase the receptive field by inserting holes into the convolution kernels. Convolution layers are not effective at capturing features from distant spatial locations, so contextual attention layers are introduced to identify where to obtain feature information from the background and to generate the missing area. First, convolution is used to compute the matching score of foreground patches with background patches (as convolutional filters). Then, a softmax layer is applied to compare and obtain attention score for each pixel. Finally, foreground patches are reconstructed with background patches by performing deconvolution on the attention score. The contextual attention layer is differentiable and fully convolutional. This network structure effectively utilizes the contextual features of the background area to reconstruct the masked area, ensuring that the final image retains only the contextual features of the background area.

For the predictions of the teacher network, if the lesion area is accurately located, the extracted contextual features after the mask operation will only contain features of the background area and not lesion features. Thus, the reconstructed image will also not contain lesion features, but only features of the non-lesion area. If the lesion area is not accurately located, the mask will not completely cover the lesion area, and the extracted contextual features will include both non-lesion area features and lesion area features. The final reconstructed image will not completely eliminate lesion features. In other words, the image reconstructed based on contextual features can restore the lesion area without introducing new features. If the predictions from the teacher network are ideal, the final reconstructed image will not contain any lesion features, achieving the goal of disrupting lesion features. Otherwise, the final reconstructed image will still contain lesion features, indicating incomplete reconstruction of lesion features.

Through lesion-aware reconstruction based on the CAtt module, we obtain the original image and the lesion-aware reconstructed image, which form a pair of samples used as input

for the subsequent scoring module. As shown in Figure 1, the precisely reconstructed image no longer exhibits the typical appearance of lesions. Instead, it resembles the texture of the surrounding healthy tissue. This approach ensures the destruction of lesion features without introducing unrelated features, laying a solid foundation for subsequent quality scoring.

### 3.3. Contrastive Learning-based Scoring Module

The scoring module is designed to evaluate the accuracy of the teacher network's predictions. Its core principle states: when the teacher network's predictions are accurate, the lesion-aware reconstruction network can completely reconstruct the lesion region features to a healthy image, while minimizing damage to non-lesion region features. This difference between original image and reconstructed image is quantified by the mapping distance in feature space - that is, the accuracy of lesion-aware reconstruction is positively correlated with the prediction accuracy of the teacher network. Consequently, the evaluation metrics of lesion-aware reconstruction can serve as detection result assessment indicators, exhibiting properties similar to the IoU metric.

To achieve this objective and obtain quantifiable metrics for lesion-aware reconstruction accuracy, this section designs a contrastive learning-based scoring network. The core concept involves bringing the representations of samples containing lesions closer together in the embedding space, while pushing them away from representations of non-lesion samples. The contrastive loss function used for model training is $\mathcal{L}_{InfoNCE}$ (Noise Contrastive Estimation):

$$\mathcal{L}_{InfoNCE} = -\log\left(\frac{\exp\left(\mathrm{sim}(z_i, z_j)/\tau\right)}{\sum_{k=1}^{N}\exp\left(\mathrm{sim}(z_i, z_k)/\tau\right)}\right) \tag{3}$$

where $z_i$ and $z_j$ are the representations of the positive sample pair, $z_k$ is the representation of the negative sample pair, $\tau$ is the temperature parameter, and sim is the similarity function.

Training based on contrastive learning enables the network to effectively learn feature representations of lesions, thereby capturing the differences in feature space before and after image reconstruction. This reflects the effectiveness of the reconstruction. To quantify this capability of distinguishing feature differences, a linear layer and a sigmoid operation are added to the final layer of the contrastive network, converting the output into a score that indicates the performance of the reconstruction. A score closer to "0" suggests that the input image slice contains almost no lesion features, meaning lesion-aware reconstruction was highly effective. Conversely, a score closer to "1" indicates that input image slice contains prominent lesion features.

Additionally, as discussed earlier, this output score can also reflect the performance of the detection network, exhibiting IoU-like properties. Leveraging this characteristic, the scoring network can effectively evaluate predictions of the teacher network. In the absence of labeled data to compute IoU, this scoring metric provides an objective assessment of prediction quality. Consequently, it allows for selection of high-quality predictions as training samples for the student network while mitigating the impact of the teacher network's bias in mandibular lesion detection.

## 4. Experiments and Results

### 4.1. Dataset

We validated the effectiveness of the principal network using a jaw lesion dataset. The dataset in this study comes from the Radiology Department of Peking University School and has received approval from the ethics review board (Approval Number: 2023-City Nature-43). Pathological and radiological diagnoses were used as ground truth. The dataset was annotated and verified by one team comprising three radiologists with over five years of experience, one radiologist with over ten years of experience, and one senior radiologist. In total, we have collected three common types of jaw lesions: cystic lesions, solid lesions, and mixed lesions, some example images are shown in Figure 2.

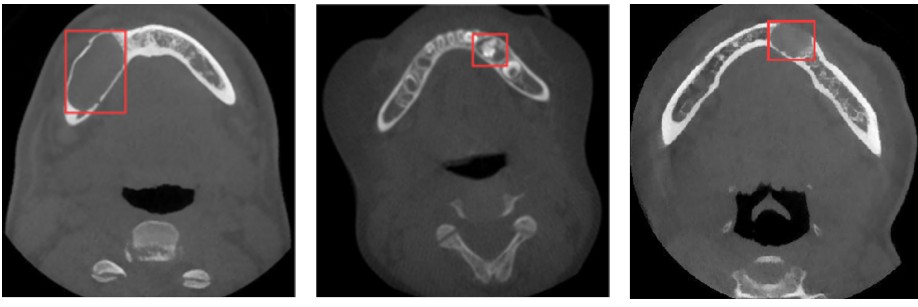

Figure 2: Examples of CBCT images with lesion areas (red boxes) showing from left to right: cystic, solid, and mixed lesion.

### 4.2. Experiments and Analysis of Lesion-aware Reconstruction Network

Healthy mandibular image slices were used as the training set for the lesion-aware reconstruction network to facilitate better learning of imaging features of healthy mandibles. Irregular masks were randomly generated for training. During training process, total variation loss is used for both the coarse and fine generation stages.

$$\mathcal{L}_{TV} = \sum_{i,j} \left( |y_{i+1,j} - y_{i,j}| + |y_{i,j+1} - y_{i,j}| \right) \tag{4}$$

where $y$ represents the normalized pixel value.

Additionally, we conducted tests on the trained lesion-aware reconstruction network to evaluate its performance in reconstructing lesion region features. Coarse reconstructed results and final reconstructed slices are shown in Figure 3. As can be observed, the lesion-aware reconstruction model based on contextual attention successfully removes the original lesion features while maximally preserving non-lesion features. By leveraging the characteristic information from healthy background mandibular images, the network generates new mandibular image slices that effectively eliminate pathological patterns.

To better visualize these lesion features, this study focuses on examining distinction between lesion and reconstructed features in the feature space. For this purpose, we take

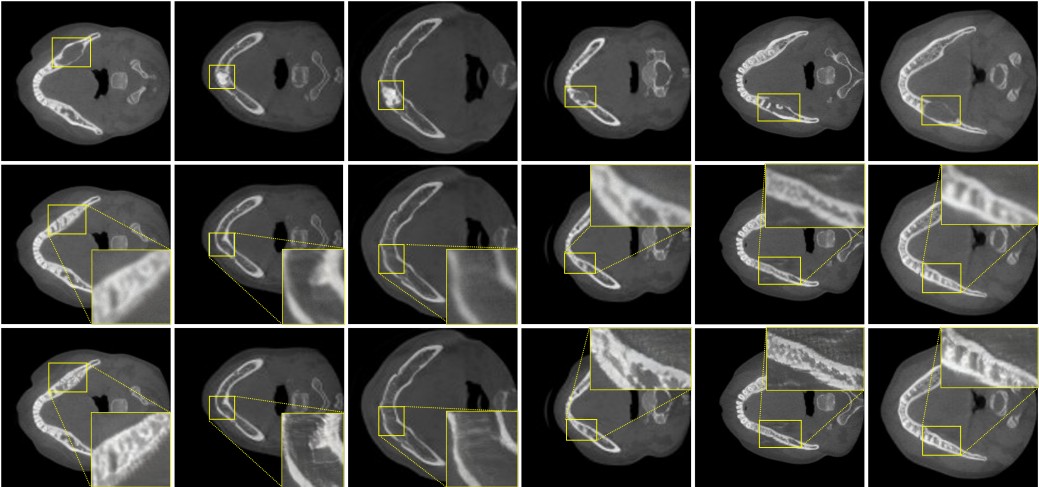

Figure 3: Feature-destroyed image slices, first row: original lesion images; second and third rows: coarse results and fine results from lesion-aware reconstruction; fourth and fifth rows: final lesion-aware reconstructed slices refined through contextual attention.

pre-reconstruction and post-reconstruction images as paired samples and feed them into the feature extraction layer of the target detection network. Then four dimensionality reduction techniques are employed to map the high-dimensional features into lower-dimensional spaces (2D and 3D) for visualization: 1) Principal Component Analysis (PCA): linear dimensionality reduction method that projects high-dimensional data onto lower-dimensional space by identifying directions of maximum variance (principal components). 2) t-Distributed Stochastic Neighbor Embedding (t-SNE): nonlinear approach that preserves local similarities between data points when mapping to lower dimensions. 3) Uniform Manifold Approximation and Projection (UMAP): manifold-based nonlinear technique that approximates high-dimensional manifold structure before projecting it into lower-dimensional space. 4) Spectral Embedding (SE): graph-theory-based method that performs dimensionality reduction through eigen decomposition of the graph Laplacian matrix. The final visualization results are shown in Figure 4. The analysis reveals that for the same lesion slice, the pre- and post-reconstruction images exhibit measurable separation in high-dimensional feature space - this separation distance corresponds to lesion features. Notably, more complete reconstruction of lesion features leads to greater separation distances, with the processed images retaining only non-lesion characteristics.

### 4.3. Experiments and Analysis of Contrastive Learning-based Scoring Module

This study conducted a series of comparative experiments, with results presented as follows: Table 1 shows performance of the student network when increasing amount of unlabeled data while keeping that of labeled data fixed. Experiments demonstrate that, under a constant labeled data volume, the student network's performance exhibits a continuous

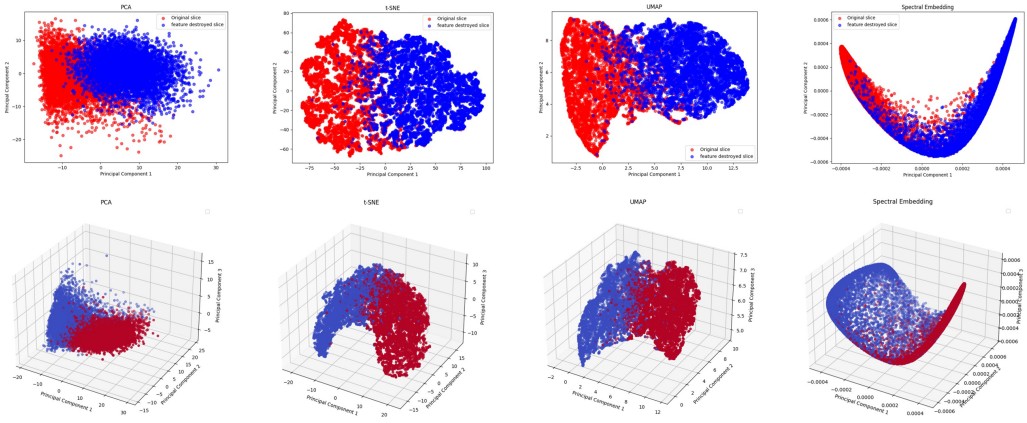

Figure 4: Visualization of lesion features: 2D in the first row and 3D in the second row. Red represents the high-dimensional features of original images, blue represents the high-dimensional features of reconstruction images.

improvement trend as more unlabeled training samples are incorporated into the training set. Table 2 presents final performance of the student network when progressively reducing proportion of labeled data while maintaining total annotation quantity (i.e., redistributing labels across fewer samples with higher density). These results indicate that even with reduced label ratios, the student network maintains competitive performance as long as total annotation budget remains unchanged. Meanwhile, to validate the effectiveness of the principal network, we conducted experiments using the same dataset split but without the principal network, using an IoU threshold (threshold of 0.5). The experimental results are shown in Table 3.

Table 1: Performance with amount of labeled data unchanged and unlabeled data increasing

| Metrics | 1:0 (Fully Supervised) | 1:1 | 1:2 | 1:3 | 1:4 |
|---|---|---|---|---|---|
| Precision | 0.793 | 0.805 | 0.814 | 0.823 | 0.828 |
| Recall | 0.664 | 0.683 | 0.697 | 0.705 | 0.717 |
| F1-score | 0.723 | 0.739 | 0.751 | 0.759 | 0.769 |
| mAP@0.5 | 0.694 | 0.725 | 0.742 | 0.758 | 0.767 |
| mAP@[.5: .95] | 0.506 | 0.539 | 0.578 | 0.604 | 0.627 |

To further verify the effectiveness of the proposed SSL strategy in mandibular lesion detection, we compared the performance of different SSL strategies on the same dataset using the mAP@0.5 metric (as shown in Table 4). We selected STAC (SOHN et al., 2020), Unbiased Teacher (LIU et al., 2021), and Soft Teacher (XU et al., 2021) as baseline methods (ZHU and CHEN, 2024), and chose Residual CycleGAN (MATSUI et al., 2025) as a

Table 2: Performance with total amount of data unchanged and proportion of unlabeled data increasing

| Metrics | 1:0 | 1:1 | 1:2 | 1:3 | 1:4 |
|---|---|---|---|---|---|
| Precision | 0.887 | 0.873 | 0.866 | 0.842 | 0.828 |
| Recall | 0.811 | 0.765 | 0.741 | 0.730 | 0.717 |
| F1-score | 0.847 | 0.815 | 0.799 | 0.782 | 0.769 |
| mAP@0.5 | 0.848 | 0.815 | 0.791 | 0.776 | 0.767 |
| mAP@[.5:.95] | 0.706 | 0.671 | 0.655 | 0.634 | 0.627 |

Table 3: Performance without the principal network (total amount of data unchanged)

| Metrics | 1:0 | 1:1 | 1:2 | 1:3 | 1:4 |
|---|---|---|---|---|---|
| Precision | 0.688 | 0.645 | 0.627 | 0.611 | 0.598 |
| Recall | 0.747 | 0.734 | 0.706 | 0.697 | 0.686 |
| F1-score | 0.737 | 0.692 | 0.622 | 0.584 | 0.547 |
| mAP@0.5 | 0.792 | 0.738 | 0.721 | 0.715 | 0.699 |
| mAP@[.5:.95] | 0.606 | 0.576 | 0.536 | 0.503 | 0.476 |

Table 4: Performance comparison of different SSL strategies in different ratios of quantities of labeled and unlabeled data

| Methods | 1:4 | 1:3 | 1:2 | 1:1 |
|---|---|---|---|---|
| STAC (SOHN et al., 2020) | 0.693 | 0.715 | 0.743 | 0.758 |
| Unbiased Teacher (LIU et al., 2021) | 0.725 | 0.741 | 0.752 | 0.764 |
| Soft Teacher (XU et al., 2021) | 0.712 | 0.733 | 0.756 | 0.784 |
| Residual CycleGAN (MATSUI et al., 2025) | 0.724 | 0.746 | 0.773 | 0.800 |
| Ours | 0.767 | 0.776 | 0.791 | 0.815 |

more recent approach. The experimental results demonstrate that compared to other methods, the SSL training strategy proposed in this paper shows significant advantages under conditions of high unlabeled data ratios. Given that clinical medical mandibular data is characterized by scarce labeled samples but abundant unlabeled data, this finding fully confirms the clinical application value of the proposed algorithm.

## 5. Limitations

First, the efficacy of the lesion-aware reconstruction module is contingent upon the availability of sufficient local healthy contextual information. When the proportion of lesioned regions becomes excessively large, both the reconstruction quality and the reliability of evaluation metrics may be compromised. Second, the selection of threshold values for pseudo-label filtering is inherently constrained by the heterogeneous imaging characteristics across different lesion types, rendering a unified thresholding approach challenging. Consequently, static thresholds may introduce systematic bias into the evaluation process.

## 6. Conclusion

Our TPS framework offers a robust solution for SSL lesion detection by addressing the critical issue of high-confidence erroneous pseudo-labels. The principal network, through its lesion-aware reconstruction and scoring mechanisms, ensures the quality of pseudo-labels, leading to more accurate and reliable detection outcomes. This model enhances the overall performance of SSL in medical image analysis, providing a valuable tool for clinical diagnosis. Future work may focus on refining the lesion-aware reconstruction and exploring its applicability to other medical imaging domains.

## Acknowledgments

This work was supported by the Beijing Natural Science Foundation under Grant L232029 and the Capital's Fund for Health Improvement and Research (CFH 2024-4-4107) from the Beijing Municipal Health Commission. (Corresponding author: Jupeng Li.)

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

## Appendix A. Experiments Settings

All experiments were conducted on a workstation equipped with 2 NVIDIA RTX 6000 Ada GPUs (48 GB VRAM each). The software environment was Ubuntu 20.04 LTS with PyTorch 1.12.1 and CUDA 11.6. For lesion detection, we used the Adaptive Cross-View Feature Mining detection network (WANG et al., 2024). The detection network was trained with an initial learning rate of 0.001 (decay factor 0.1) using SGD (momentum 0.9, weight decay 0.0005). For lesion-aware reconstruction, we adopted a Context Attention GAN, trained with a learning rate of 0.0001 using the Adam optimizer ($\beta_1 = 0.5$, $\beta_2 = 0.999$). For the contrastive learning network, the initial learning rate was set to 0.001, and the temperature parameter was set to 0.1.

## Appendix B. Experiments and Analysis of Contrastive Learning-based Scoring Module

We first validate the IoU-like characteristics of the scoring network. Based on the existing annotated bounding boxes, we randomly generated detection boxes with different IoU values. The generated bounding boxes were passed through the scoring network to obtain corresponding scores. Multiple experiments were conducted in this part, and the mean scores from the scoring network corresponding to different IoU bounding boxes across multiple experiments are shown in Table 5. The curve plot of the average experimental scores

Table 5: Different IoU detection boxes under different experiments correspond to the mean output of the scoring network

| IoU Range | 1 | 2 | 3 | 4 | 5 | 6 | 7 | 8 |
|---|---|---|---|---|---|---|---|---|
| (0, 0.1] | 0.05 | 0.02 | 0.05 | 0.04 | 0.05 | 0.02 | 0.03 | 0.05 |
| (0.1, 0.2] | 0.07 | 0.05 | 0.07 | 0.05 | 0.09 | 0.05 | 0.08 | 0.08 |
| (0.2, 0.3] | 0.18 | 0.20 | 0.21 | 0.15 | 0.25 | 0.17 | 0.25 | 0.22 |
| (0.3, 0.4] | 0.49 | 0.54 | 0.51 | 0.54 | 0.47 | 0.50 | 0.50 | 0.52 |
| (0.4, 0.5] | 0.52 | 0.54 | 0.51 | 0.54 | 0.55 | 0.56 | 0.51 | 0.50 |
| (0.6, 0.7] | 0.55 | 0.57 | 0.55 | 0.51 | 0.49 | 0.55 | 0.56 | 0.55 |
| (0.7, 0.8] | 0.56 | 0.58 | 0.52 | 0.56 | 0.54 | 0.58 | 0.57 | 0.57 |
| (0.8, 0.9] | 0.63 | 0.66 | 0.62 | 0.67 | 0.63 | 0.61 | 0.62 | 0.65 |
| (0.9, 1.0] | 0.79 | 0.82 | 0.82 | 0.83 | 0.80 | 0.78 | 0.82 | 0.81 |

versus the average generated IoU values is presented in Figure 5. To more directly compare the correlation between IoU values and scoring network outputs, we employed Pearson correlation coefficient (Pearson $r$) and Spearman rank correlation coefficient (Spearman $\rho$) along with their corresponding $p$-values to quantify the IoU metric and the output scores from the scoring network. The obtained similarity results are shown in Table 6. It is evident that the network scores exhibit very high correlation with IoU metrics, while the $p$-values approach zero, indicating that the scoring network can objectively evaluate the prediction quality of bounding boxes. This score represents an IoU-like metric value that effectively represents the prediction performance of the detection network.

Second, we conducted experiments and analysis on the scoring network threshold selection. We performed experiments on fixed score, fixed percentage, and adaptive threshold methods respectively, with experimental results shown in Table 7. The adaptive threshold method, which provides threshold scores based on the overall performance of labeled data, outperformed both the fixed score and fixed percentage methods. For lesion detection, a unified threshold is difficult to adapt to all lesion categories. Therefore, we applied category-specific adaptive threshold methods for different lesion types, with the final experimental results shown in Table 8.

Table 6: The correlation between scoring network score and IoU of different experiments

| Experiments | Pearson $r$ | Pearson $p$ | Spearman $\rho$ | Spearman $p$ |
|---|---|---|---|---|
| 1 | 0.954 | 1.78e-05 | 0.964 | 7.32e-06 |
| 2 | 0.945 | 2.13e-05 | 0.971 | 6.16e-06 |
| 3 | 0.947 | 1.92e-05 | 0.945 | 1.44e-05 |
| 4 | 0.941 | 2.44e-05 | 0.987 | 5.06e-06 |
| 5 | 0.966 | 1.23e-05 | 0.953 | 9.47e-06 |
| 6 | 0.959 | 1.88e-05 | 0.955 | 8.03e-06 |
| 7 | 0.940 | 2.66e-05 | 0.949 | 1.21e-05 |
| 8 | 0.945 | 2.04e-05 | 0.951 | 1.06e-05 |

Table 7: Network performance under different threshold selection policies

| Threshold Selections | Score | Percentage | Quantity After Filtering | Student Network mAP@0.5 |
|---|---|---|---|---|
| Fixed Score | 0.8 | 56.77% | 560 | 85.06% |
| Fixed Score | 0.7 | 71.25% | 703 | 85.29% |
| Fixed Score | 0.6 | 83.14% | 821 | 85.42% |
| Fixed Score | 0.5 | 84.90% | 838 | 85.39% |
| Fixed Percentage | 0.826 | 50% | 494 | 84.94% |
| Fixed Percentage | 0.787 | 60% | 592 | 85.13% |
| Fixed Percentage | 0.715 | 70% | 691 | 85.26% |
| Fixed Percentage | 0.634 | 80% | 790 | 85.77% |
| Adaptive Threshold | 0.833 | 46.15% | 455 | 84.97% |
| Adaptive Threshold | 0.793 | 57.99% | 572 | 85.01% |
| Adaptive Threshold | 0.741 | 65.41% | 646 | 85.31% |
| Adaptive Threshold | 0.700 | 71.23% | 703 | 85.46% |
| Adaptive Threshold | 0.633 | 79.58% | 786 | **85.81%** |

Table 8: The final network performance for the respective category under the adaptive threshold method

| Lesion Types | Score | mAP@0.5 | mAP@[.5:.95] |
|---|---|---|---|
| Cystic | 0.734 | 0.939 | 0.760 |
| Solid | 0.689 | 0.856 | 0.742 |
| Mixed | 0.541 | 0.787 | 0.639 |
| Average | – | 0.861 | 0.714 |

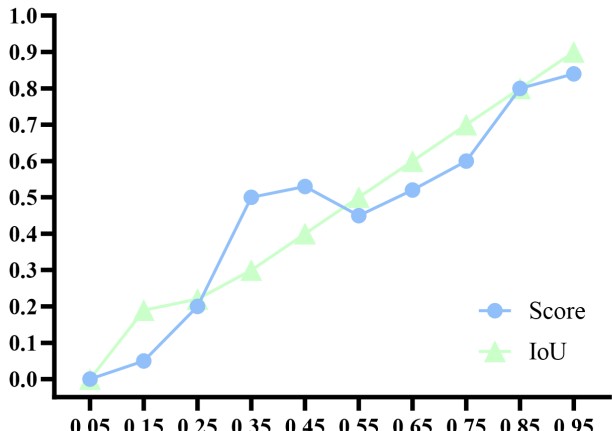

Figure 5: The mean scores and mean generated IoU values

