# OpenReview forum: "Lesion-Aware Reconstruction with Principal Network: Enhancing Pseudo-Label Reliability in Semi-Supervised Clinical Lesion Detection"
_MIDL.io/2026/Conference — MIDL 2026 Poster_

### Official Review · Reviewer_tBuM · 2025-12-21

**Confidence:** 4
**Preliminary Rating:** 2

**Summary:**

This paper addresses the problem of unreliable pseudo labels in semi supervised lesion detection, with a particular focus on jaw lesion detection in CBCT images. The authors propose a Teacher Principal Student framework, where an additional Principal network is introduced to evaluate the quality of pseudo labels generated by the teacher model. The key idea is to assess pseudo label reliability through lesion aware reconstruction, where predicted lesion regions are masked and reconstructed using contextual attention, and the reconstruction effect is quantified via a contrastive learning based scoring module. This score is then used to filter low quality pseudo labels before training the student detector. Experiments on a jaw lesion dataset demonstrate improved detection performance over several semi supervised baselines under different labeled to unlabeled data ratios, and the authors report a reduced gap between semi supervised and fully supervised settings. Overall, the paper aims to improve pseudo label reliability rather than proposing a new detection architecture, and its significance lies in refining the training strategy for semi supervised clinical lesion detection.

**Strengths:**

1. The paper tackles a practically relevant problem in semi supervised medical image analysis, namely the negative impact of high confidence but incorrect pseudo labels in teacher student frameworks, which is well motivated in the context of limited clinical annotations.

2. The introduction of a dedicated Principal network to explicitly assess pseudo label quality represents a clear architectural separation between prediction generation and reliability estimation, which improves conceptual clarity compared to standard self training pipelines.

3. The lesion aware reconstruction mechanism leverages anatomical context and reconstruction consistency, providing an intuitive and clinically interpretable criterion for evaluating whether predicted lesion regions are meaningful.

4. The experimental protocol considers multiple labeled to unlabeled data ratios, and the reported results consistently show improvements over commonly used semi supervised detection methods, suggesting that the approach has practical potential under annotation constrained settings.

5. The paper is generally well structured, with a clear problem statement, method description, and experimental analysis, and the writing is mostly understandable for readers familiar with semi supervised learning in medical imaging.

**Weaknesses:**

1. The methodological novelty is limited, as the proposed framework mainly combines existing components including teacher student learning, contextual attention based image inpainting, and contrastive learning based scoring, without introducing a fundamentally new learning principle.

2. The core assumption that lesion aware reconstruction quality is a reliable proxy for pseudo label correctness is not sufficiently justified theoretically, and the connection between reconstruction quality and detection accuracy remains largely heuristic.

3. The Principal network introduces substantial additional complexity, including a reconstruction model and a contrastive scorer, but the paper does not provide a clear analysis of computational overhead, training stability, or scalability to larger datasets or higher resolution images.

4. The evaluation is restricted to a single dataset and a single anatomical site, which limits the generalizability of the conclusions. It remains unclear whether the proposed strategy would transfer to other lesion types, imaging modalities, or detection tasks.

5. Comparisons are limited to a small set of semi supervised baselines, and more recent or stronger detection oriented SSL methods are not included, making it difficult to fully assess the competitiveness of the approach.

6. Several design choices, such as the thresholding strategy for selecting high quality pseudo labels based on the scoring output, appear ad hoc and are not systematically analyzed or justified.

7. The reported performance gains, while consistent, are relatively modest, and it is unclear whether the added architectural and training complexity is justified by the observed improvements.

**Detailed Comments:**

The paper would benefit from clearer quantitative analysis of how reconstruction scores correlate with true detection quality, for example by explicitly reporting correlations with IoU where labels are available. Additional ablation studies isolating the impact of reconstruction and contrastive scoring would improve clarity. The description of the Principal network training procedure could be made more precise, particularly regarding data splits and supervision signals. Some figures illustrating failure cases would help readers understand the limitations of the proposed approach.

**Justification Of The Preliminary Rating:**

While the paper addresses an important and relevant problem in semi supervised lesion detection, the overall contribution is incremental rather than transformative. The proposed framework mainly integrates existing techniques in a new combination, and the empirical validation is limited in scope. The added complexity of the Principal network is not fully justified by the scale of performance gains, and the lack of evaluation across multiple datasets or modalities weakens the general impact of the work. As a result, although the approach shows promise and may be of interest to a narrow application domain, it does not yet meet the level of novelty, generality, and experimental rigor expected for acceptance, leading to a weak reject recommendation.

**Questions To Address In The Rebuttal:**

How robust is the reconstruction based scoring mechanism when lesion boundaries are ambiguous or when lesions occupy a very small area? Can the authors provide evidence that the proposed method generalizes to other datasets or lesion types? How does the computational cost of the full TPS framework compare to standard teacher student baselines? Would similar gains be achievable with simpler uncertainty or consistency based filtering strategies?

---

> ### Author Response · Authors · 2026-01-25
>
> ```markdown
> ## Question 1
> **How robust is the reconstruction-based scoring mechanism when lesion boundaries are ambiguous or when lesions occupy a very small area?**
>
> ### Response
> Ambiguous boundaries and very small lesion areas are core difficulties in medical image detection and are also major sources of pseudo-label noise. In our dataset, the included lesion types—especially **mixed lesions**—provide an effective testbed for evaluating robustness. Mixed lesions (cystic–solid mixtures) typically present **ambiguous boundaries** and atypical internal features, while small cystic/solid lesions represent the challenge of **tiny target regions**.
>
> Our method shows strong capability under these conditions. For ambiguous boundaries, the scoring mechanism tends to assign **lower scores**, and the use of **adaptive (or lower) thresholds** provides flexibility in filtering unreliable pseudo-labels; this aligns with the largest performance gains observed on mixed lesions. For small lesions, as long as they are initially proposed by the detector, our mechanism can still evaluate the associated feature changes (supported by the results on solid lesions). Overall, on our dataset, our method achieves superior performance compared with baseline models.
>
> ## Question 2
> **Can the authors provide evidence that the proposed method generalizes to other datasets or lesion types?**
>
> ### Response
> The proposed method was designed to be applicable to different tissues and a wide range of disease types, although our current experiments are mainly conducted on an oral CBCT dataset. This dataset contains **cystic, solid, and mixed lesions** and includes challenging radiological characteristics, such as **ill-defined/wide transitional zones**, **large scale variations**, and **heterogeneous internal density/texture**.
>
> The strong performance on this dataset suggests robustness to unclear boundaries and scale variations, indicating potential generalization to other datasets with similar imaging challenges. We plan to conduct experiments on additional datasets in future work to further validate cross-dataset generalization.
>
> ## Question 3
> **How does the computational cost of the full TPS framework compare to standard teacher-student baselines?**
>
> ### Response
> The Principal Network contains two components: the **Lesion-Aware Reconstruction module** and the **contrastive scoring module**. To manage training costs, we use a pre-trained reconstruction model and fine-tune it on healthy images. After this stage, both the reconstruction and scoring modules are integrated into the teacher-student framework, and their parameters are **frozen**, so they do not require additional optimization during subsequent semi-supervised learning iterations.
>
> During inference, these modules are **not involved**, resulting in **no additional inference overhead**. Therefore, while the initial pre-training/fine-tuning stage requires computation, the operational cost of the full TPS framework during the main SSL training and inference phases is comparable to standard teacher-student baselines, since the key addition is a frozen evaluation network.
>
> ## Question 4
> **Would similar gains be achievable with simpler uncertainty- or consistency-based filtering strategies?**
>
> ### Response
> Uncertainty- or consistency-based filtering strategies are attractive due to their simplicity, but in semi-supervised medical image detection they face inherent limitations. For early-stage lesions, blurred boundaries, or high inter-class similarity, models may produce **incorrect yet highly confident predictions**. In such cases, simple confidence-thresholding or uncertainty estimation derived from the model itself may fail to identify these “confident errors.”
>
> Our Principal Network is designed to address this limitation by acting as an independent **evaluator**, rather than a self-referential filter. Through lesion-aware reconstruction, we transform the region implied by the teacher’s pseudo-label into a “healthy” version, and then assess reliability using a score with **IoU-like characteristics**. This design provides a more objective basis for pseudo-label quality assessment than purely model-internal confidence/uncertainty signals.
> ```

---

> > ### Comment · Area_Chair_6jF4 · 2026-02-02
> >
> > Dear Reviewer,
> > Please indicate whether the authors’ responses have resolved your questions and provide your final rating with a justification. Thanks!

---

> ### Author Response · Authors · 2026-02-03
>
> We sincerely appreciate your review and valuable feedback on our work.
> To address the issue of unreliable pseudo-labels in semi-supervised lesion detection, we introduce an independent principal network to enhance the reliability of pseudo-labels generated by the teacher network. The primary motivation for proposing the principal network is to overcome the limitations of existing filtering strategies based solely on uncertainty or consistency metrics. Our method is validated on a jaw lesion dataset characterized by blurred boundaries, atypical features, and significant variations in lesion area, demonstrating its robustness. The Lesion-Aware Reconstruction module reduces computational costs through pre-training and fine-tuning, thus requiring no additional training.  Thank you.

---

### Official Review · Reviewer_NV96 · 2026-01-10

**Confidence:** 4
**Preliminary Rating:** 4
**Final Rating:** 4

**Summary:**

This manuscript proposes a novel Teacher–Principal–Student (TPS) semi-supervised learning framework for clinical lesion detection. The proposed method is motivated by the issue of error amplification caused by high-confidence but incorrect pseudo-labels in teacher–student SSL pipelines. The key contribution is the introduction of a Principal Network, which evaluates pseudo-label reliability through lesion-aware reconstruction and contrastive scoring. By reconstructing teacher-predicted lesion regions into healthy-appearing tissue using contextual attention–based inpainting, and measuring feature-space discrepancies between original and reconstructed images, the method filters out low-quality pseudo-labels before student training. Experiments on a jaw lesion CBCT dataset demonstrate consistent improvements over standard SSL baselines. The proposed method reportedly narrows the gap between semi-supervised and fully supervised performance.

**Strengths:**

The paper identifies a genuine and important limitation of SSL in medical imaging: the propagation of high-confidence teacher errors..

Introducing a third “principal” module to independently assess pseudo-label quality is an interesting extension beyond conventional TS frameworks.

Repurposing contextual-attention inpainting as a mechanism to destroy lesion features for quality assessment is creative and aligns well with clinical intuition.

Jaw lesion detection from CBCT images is a meaningful real-world task with scarce annotations, making SSL particularly appropriate.

The reported improvements are consistent across experimental settings.

**Weaknesses:**

The claim that the contrastive score exhibits “IoU-like properties” is intuitive but not formally justified or empirically validated.

The respective contributions of Lesion-aware reconstruction, Contrastive learning–based scoring, Final sigmoid scoring head are not clearly disentangled.

The reconstruction assumption (lesion → healthy tissue) may not hold for other modalities or diffuse pathologies. No cross-dataset or cross-organ validation is provided.

Key information is missing or unclear, including:

Adding a reconstruction network and contrastive scorer likely increases training cost, but runtime, memory usage, or training stability are not reported.

The manuscript contains frequent grammatical errors, awkward phrasing, and inconsistent terminology (e.g., “lesion-aware” vs. “lesion-aware, reconstruction”), which affect readability.

**Detailed Comments:**

It is unclear whether the Principal Network score correlates with ground-truth IoU or labeled data.

No evidence demonstrates that this method  generalizes to other lesion types, organs, or imaging modalities.

The additional computational cost introduced by the Principal Network is unclear.

**Justification Of Final Rating:**

I have concerns about the manuscript that it lacks theoretical grounding, ablation analysis, and clear reporting of dataset and training details. However, this paper in general presents a novel extension to semi-supervised lesion detection by introducing a reconstruction-based pseudo-label quality assessor. The experimental results are promising and the gains over existing SSL baselines are consistent.

**Justification Of The Preliminary Rating:**

This paper presents a novel extension to semi-supervised lesion detection by introducing a reconstruction-based pseudo-label quality assessor. The idea is creative, empirically effective, and well-aligned with clinical intuition. While the experimental results are promising and the gains over existing SSL baselines are consistent, the manuscript currently lacks deeper theoretical grounding, stronger ablation analysis, and clearer reporting of dataset and training details.

**Questions To Address In The Rebuttal:**

How strongly does the Principal Network score correlate with ground-truth IoU on labeled data?

What happens if lesion-aware reconstruction is replaced with simpler perturbations (e.g., random masking or blurring)?

Can the method generalize to other lesion types, organs, or imaging modalities?

What is the additional computational cost introduced by the Principal Network?

How sensitive is performance to the scoring threshold or temperature parameter τ?

Are there failure cases where reconstruction removes clinically relevant subtle patterns?

---

> ### Author Response · Authors · 2026-01-25
>
> ```markdown
> We thank the reviewer for the comprehensive and insightful questions, which helped us clarify and strengthen our methodology. Below are our point-by-point responses.
>
> ## Question 1
> How strongly does the Principal Network score correlate with ground-truth IoU on labeled data?
>
> ### Response
> We agree that empirically validating the “IoU-like property” of our score is essential. In the revised manuscript, we added **Appendix B: Experiments and analysis of contrastive scorer network**. On labeled data, we generate bounding boxes with varying IoU values via bounding-box perturbation, obtain their corresponding scores from the contrastive scorer, and analyze the score–IoU relationship across multiple trials using **Pearson correlation coefficient (PCC)** and **Spearman rank correlation coefficient (SRCC)** (with p-values). Across **8 independent trials**, results are consistent: average **PCC ≈ 0.950** and **SRCC ≈ 0.959**, with all correlation p-values **< $1\times10^{-5}$**. Supplementary scatter plots further illustrate an approximately linear positive trend. These results support that the proposed score is a reliable indicator of pseudo-label quality.
>
> ## Question 2
> What happens if lesion-aware reconstruction is replaced with simpler perturbations (e.g., random masking or blurring)?
>
> ### Response
> We added an ablation study in the revised experimental section (**Section 4.3**) to compare different perturbation strategies within the teacher-student framework (e.g., random masking/blurring versus our lesion-aware reconstruction). The results show that introducing the **Principal Network with lesion-aware reconstruction** achieves better overall performance than simpler perturbation-based alternatives, indicating that reconstruction contributes meaningfully to improving pseudo-label reliability.
>
> ## Question 3
> Can the method generalize to other lesion types, organs, or imaging modalities?
>
> ### Response
> While our current experiments focus on an oral CBCT dataset, the method is designed to be broadly applicable. The dataset includes **cystic, solid, and mixed lesions** and exhibits challenging imaging characteristics such as ill-defined boundaries, large scale variation, and heterogeneous internal texture. Strong performance under these conditions suggests robustness to unclear boundaries and appearance variation, which are common in many medical imaging settings. We plan to validate cross-organ and cross-modality generalization on additional datasets in future work.
>
> ## Question 4
> What is the additional computational cost introduced by the Principal Network?
>
> ### Response
> The Principal Network consists of the **Lesion-Aware Reconstruction module** and the **contrastive scorer**. The reconstruction model is first **pre-trained/fine-tuned** on healthy images. After this stage, both the reconstruction and scoring modules are integrated into the teacher-student framework with their parameters **frozen** during semi-supervised iterations, and they are **not used during inference**. Therefore, there is **no additional inference overhead**, and the main extra cost is the initial pre-training/fine-tuning stage.
>
> ## Question 5
> How sensitive is performance to the scoring threshold or temperature parameter $\tau$?
>
> ### Response
> We investigated three threshold-selection strategies: **fixed score threshold**, **percentile-based threshold**, and **adaptive threshold** (details reported in **Appendix B**). Performance varies only slightly across reasonable thresholds (e.g., fixed thresholds 0.5–0.8 yield mAP@0.5 around **85.06%–85.42%**, and the adaptive method reaches **85.81%**), indicating the overall performance is not highly sensitive to threshold choice. In contrast, the temperature parameter $\tau$ in InfoNCE controls similarity sharpness (smaller $\tau$ enhances discrimination; larger $\tau$ smoothens distributions) and is a more sensitive hyperparameter that requires tuning.
>
> ## Question 6
> Are there failure cases where reconstruction removes clinically relevant subtle patterns?
>
> ### Response
> Our approach is not intended to detect every subtle deviation from normal appearance. Instead, it assesses whether a **salient abnormal region** proposed by the detector is sufficiently anomalous, serving as a precision-oriented pseudo-label filter. As a result, extremely subtle patterns may be reconstructed as healthy, which may reduce sensitivity to very subtle findings. We acknowledge this limitation and consider it an important direction for future improvement and broader clinical validation.
> ```

---

### Official Review · Reviewer_XMsG · 2026-01-10

**Confidence:** 3
**Preliminary Rating:** 3
**Final Rating:** 4

**Summary:**

This manuscript proposes a newly designed semi-supervised framework for the lesion detection task, using a lesion-aware reconstruction module and contrastive learning with gated control. A jaw lesion dataset is used for performance validation. The experimental design initially validates the efficacy of the proposed method.

**Strengths:**

+ The topic of this manuscript is timely and interesting.
+ The design of the lesion-aware reconstruction module is sound. The experimental design initially validates the efficacy of the proposed method.
+ The authors provide the link to the source code.

**Weaknesses:**

- Since the proposed method involves image generation, and an object detection task typically includes a larger proportion of non-lesion regions (i.e., bounding box) compared to a segmentation task (i.e., exact pixel mask), the lesion-aware reconstruction module may benefit from this characteristic and thus produce better results. If the task were changed to a segmentation problem, would the proposed method face certain limitations? The authors are appreciated for discussing the feasibility and potential constraints of extending the method to segmentation tasks.

- The performance comparisons in the experimental design are relatively basic. The authors are expected to include proper references for the competing methods in Table 3. If these methods are intended merely as baselines, this should be clearly stated.

- The training environment settings seem missing, such as the detection framework used, GPU specifications, and other relevant implementation details.

- A dedicated Limitations section would be welcome.

**Detailed Comments:**

- The authors are expected to spell out the full names corresponding to P and R in Tables 1 and 2.

- In Figure 1, should "Principle Network" be corrected to “Principal Network”?

**Justification Of Final Rating:**

Thanks for the authors' careful and detailed response. The authors have addressed all my concerns, including major concerns and minor concerns. I have changed my rating from Borderline to Weak accept.

**Justification Of The Preliminary Rating:**

The image generation module is interesting and sound. However, experimental design is relatively weak. The overall quality is fine, even though there are some concerns. This manuscript is easy to follow.

**Questions To Address In The Rebuttal:**

Please address all concerns in Weaknesses and Detailed Comments.

---

> ### Author Response · Authors · 2026-01-25
>
> We sincerely thank the reviewer for the insightful comments and constructive suggestions, which have helped us significantly improve the manuscript. Our point-by-point responses and corresponding revisions are detailed below.
>
> ## Comment 1
> Since the proposed method involves image generation, and an object detection task typically includes a larger proportion of non-lesion regions (i.e., bounding box) compared to a segmentation task (i.e., exact pixel mask), the lesion-aware reconstruction module may benefit from this characteristic and thus produce better results. If the task were changed to a segmentation problem, would the proposed method face certain limitations? The authors are appreciated for discussing the feasibility and potential constraints of extending the method to segmentation tasks.
>
> ### Response
> The core objective of our method is to enhance the reliability of pseudo-labels generated by the teacher network within a semi-supervised learning framework. This is achieved by introducing a principal network capable of performing healthy reconstruction of lesion regions. Specifically, the Lesion-Aware Reconstruction module within the principal network reconstructs a healthy version of the input image, and a subsequent contrastive scoring module evaluates the quality of the teacher-generated pseudo-labels, thereby improving their reliability.
>
> Regarding the applicability of our method to segmentation tasks, your suggestion offers valuable insight. We plan to further explore the extensibility and adaptability of the proposed approach to segmentation tasks based on this direction.
>
> ## Comment 2
> The performance comparisons in the experimental design are relatively basic. The authors are expected to include proper references for the competing methods in Table 3. If these methods are intended merely as baselines, this should be clearly stated.
>
> ### Response
> The methods compared in Table 3 (STAC, Unbiased Teacher, and Soft Teacher) are indeed widely recognized baseline methods in semi-supervised object detection, as noted in survey literature. Zhu and Chen, in their survey article *"A Survey on Open-Vocabulary Detection and Segmentation: Past, Present, and Future"* (IEEE TPAMI, 2024), mention that the aforementioned three methods are widely recognized as representative semi-supervised object detection approaches and are frequently adopted as baselines in subsequent research. To ensure the completeness and clarity of our references, we will formally cite this literature in the revised manuscript and explicitly state in the main text that these methods serve as mainstream baselines for fair comparison.
>
> **Reference**
> [1] C. Zhu and L. Chen, “A Survey on Open-Vocabulary Detection and Segmentation: Past, Present, and Future,” *IEEE Transactions on Pattern Analysis and Machine Intelligence*, vol. 46, no. 12, pp. 8954–8975, Dec. 2024, doi: 10.1109/TPAMI.2024.3413013.
>
> ## Comment 3
> The training environment settings seem missing, such as the detection framework used, GPU specifications, and other relevant implementation details.
>
> ### Response
> We apologize for the omission. We have now added a comprehensive description of the experimental setup to ensure reproducibility. The details are as follows:
>
> - **Hardware:** 2 × NVIDIA RTX 6000 Ada GPUs ( 48GB .
> - **Software:** Ubuntu 20.04 LTS, PyTorch 1.12.1, CUDA 11.6.
> - **Detection Framework:** The Adaptive Cross-View Feature Miningdetection network was used as the backbone detector.
>
> More detailed hyperparameters and configuration details are provided in the new **Appendix A: Experiments Settings**.
>
> ## Comment 4
> A dedicated Limitations section would be welcome.
>
> ### Response
> We fully agree that a thorough discussion of limitations is crucial for academic rigor. In response, we have added a new **Section 5: Limitations** to the manuscript. The primary limitations identified for our method are:
>
> 1. The Lesion-Aware Reconstruction module may produce biased healthy reconstructions when the lesion area occupies an excessively large proportion (typically >50%) of the image, as the healthy context for reliable inpainting becomes insufficient.
> 2. The threshold selection in the contrastive scoring module currently requires task-specific tuning. For optimal performance across different lesion types with varied characteristics, an adaptive thresholding strategy would be more desirable, which is a direction for future work.
>
> ## Comment 5
> The authors are expected to spell out the full names corresponding to P and R in Tables 1 and 2.
>
> ### Response
> Using abbreviations in tables can be confusing. In the revised Tables 1 and 2, “P” has been expanded to “Precision” and “R” to “Recall” in the column headers.
>
> ## Comment 6
> In Figure 1, should "Principle Network" be corrected to “Principal Network”?
>
> ### Response
> We have thoroughly checked the manuscript and corrected all instances of “Principle Network” to the correct term, “Principal Network.” Figure 1 has been updated accordingly.

---

### Author Rebuttal · Authors · 2026-01-25

**Rebuttal:**

We sincerely thank all reviewers for their valuable time, insightful comments, and constructive suggestions. Your feedback has been instrumental in greatly enhancing the rigor, clarity, and presentation of our manuscript. We have carefully addressed each point in our official response.

**Supporting Material:**

/attachment/9763a72ff7b67ed74b6c028366b5c78a62d92158.pdf

---

### Comment · Area_Chair_6jF4 · 2026-01-26
**Start of Discussion Phase**

Thanks to the authors for providing a detailed rebuttal and for engaging constructively with the reviewers’ comments.

I kindly ask all reviewers to carefully read the rebuttal and assess whether the authors’ responses sufficiently address the raised concerns. If aspects remain unclear or require further clarification, please use the official discussion/comments function to ask follow-up questions and engage in discussion with the authors during this phase. Reviewers are also welcome to comment on and respond to the reviews and rebuttal points raised by other reviewers, where relevant.

Please note that by the end of the discussion period, all reviewers are expected to verify and, if appropriate, update their scores. Even if you agree with the authors’ responses but decide not to change your rating, please leave a brief comment indicating that you have read and considered the rebuttal.

---

### Meta-Review · Area_Chair_6jF4 · 2026-02-09

**Recommendation:** Accept (Poster)
**Confidence:** 4

**Metareview:**

Justification:
The paper addresses a relevant problem in semi-supervised medical image analysis and proposes a clear Teacher–Principal–Student framework with an intuitive lesion-aware reconstruction strategy for pseudo-label filtering. The method is technically sound and shows consistent improvements on a real-world jaw lesion CBCT dataset.
The initial reviews included one weak reject and two weak accept recommendations. The weak reject reviewer did not respond after the rebuttal; while the authors addressed the raised questions, the main weaknesses largely remain. These include the incremental nature of the contribution, heuristic assumptions, limited validation scope, modest gains relative to added complexity, and restricted baseline comparisons. Overall, this is a borderline acceptance that is best suited for a poster presentation.


**Summary:**

The paper proposes a semi-supervised learning framework for clinical lesion detection that focuses on improving pseudo-label reliability rather than modifying the detection architecture. The method introduces a Teacher–Principal–Student (TPS) framework, where an additional Principal network evaluates the quality of teacher-generated pseudo labels. Pseudo-label reliability is assessed through lesion-aware reconstruction, in which predicted lesion regions are masked and reconstructed using contextual attention, and through a contrastive learning–based scoring mechanism that measures feature discrepancies between original and reconstructed images. Low-quality pseudo labels are filtered before training the student model.

**Strengths:**
1.	Tackles a relevant and timely problem in semi-supervised medical imaging, namely unreliable pseudo-labels in teacher–student frameworks.

2.	Proposes a clear Teacher–Principal–Student framework with a dedicated module for pseudo-label quality assessment.

3.	Introduces a lesion-aware reconstruction strategy that is intuitive, clinically motivated, and technically sound.

4.	Demonstrates consistent improvements across different labeled-to-unlabeled ratios on a meaningful real-world task (jaw lesion detection in CBCT).

5.	Clearly written paper with a sound experimental design and publicly available code.


**Remaining weaknesses**

1.	The contribution is largely incremental, combining existing SSL components without introducing a fundamentally new learning principle.

2.	The core assumption that lesion-aware reconstruction reliably reflects pseudo-label correctness remains heuristic, with limited theoretical grounding and uncertain robustness in realistic failure cases.

3.	Generalizability is unclear, as evaluation is restricted to a single dataset, organ, and modality, and the limitations and failure cases are only briefly discussed.

4.	The added architectural complexity is not fully justified, as quantitative analysis of training cost, scalability, and stability is missing and performance gains are modest.

5.	Comparisons are limited to a narrow set of SSL baselines, making it difficult to fully assess competitiveness.

6.	Code usability is limited by the absence of a license and documentation, and language quality and terminology inconsistencies still affect readability.

---

### Decision · Program_Chairs · 2026-02-13

Accept (Poster)